# Ozone Effectiveness on Wheat Weevil Suppression: Preliminary Research

**DOI:** 10.3390/insects10100357

**Published:** 2019-10-18

**Authors:** Darija Lemic, Davor Jembrek, Renata Bažok, Ivana Pajač Živković

**Affiliations:** University of Zagreb, Faculty of Agriculture, Department for Agricultural Zoology, Svetošimunska 25, 10000 Zagreb, Croatia; davorjembrek@gmail.com (D.J.); rbazok@agr.hr (R.B.); ipajac@agr.hr (I.P.Z.)

**Keywords:** *Sitophilus granarius*, ozone, ozonation, mortality, efficiency, walking activity, velocity

## Abstract

Insect infestations within stored product facilities are a major concern to livestock and human food industries. Insect infestations in storage systems can result in economic losses of up to 20%. Furthermore, the presence of insects and their waste and remains in grain and stored foods may pose a health risk to humans and livestock. At present, pests in commercial storage are managed by a combination of different methods ranging from cleaning and cooling to treatment of the stored material with contact insecticides or fumigation. The availability of pesticides for the treatment of grain and other stored products is decreasing owing, in some cases, to environmental and safety concerns among consumers and society, thus emphasizing the need for alternative eco-friendly pest control methods. One of the potential methods is the use of ozone. Although the mechanism of action of ozone on insects is not completely known, the insect’s respiratory system is a likely the target of this gas. The main goal of this investigation was to determine the efficacy of ozone in the suppression of adult wheat weevils *Sitophilus granarius*. In the experiments conducted, different durations of ozone exposure were tested. In addition to ozone toxicity, the walking response and velocity of wheat weevils were investigated. The results showed the harmful effects of ozone on these insects. In addition to mortality, ozone also had negative effects on insect speed and mobility. The efficiency of the ozone treatment increased with increasing ozone exposure of insects. The ability of ozone to reduce the walking activity and velocity of treated insects is a positive feature in pest control in storage systems, thereby reducing the possibility of insects escaping from treated objects. The results of this investigation suggest that ozone has the potential to become a realistic choice for suppressing harmful insects in storage systems for humans and livestock, either alone or as a complement to other control methods.

## 1. Introduction

Insect infestation within storage facilities is a major problem in both human and livestock feed industries. Storage systems (silos, warehouses, containers) are perfect habitats for insects that feed on stored food: they are protected from weather extremes, have unrestricted access to food sources, and live undisturbed [1]. Losses due to damage caused by the stored food diet of insects and contamination with mycotoxins can exceed $500 million annually worldwide [2]. Insect infections in storage facilities can lead to economic losses of 9% in developed countries and up to 20% in developing countries. Furthermore, the presence of insects and their remains in grain and other stored food poses a threat to human and livestock health [3]. Currently, pests in storage are suppressed using a combination of different methods, from cleaning and cooling to thermal treatment of stored material with contact insecticides or fumigation. Eggs and larval developmental stages are usually not directly affected by contact insecticides. Fumigation influences these developmental stages but is also problematic because it has high toxicity and great potential for the development of resistant strains of harmful insects [4]. Previously, the standard treatment of plant material in quarantine was fumigation using methyl bromide [5]. However, the use of methyl bromide was banned in 1995 because an international convention linked this fumigant to a decrease in the ozone layer present in the Earth’s stratosphere [6]. Due to the limited possibilities of pest control in storage and transport systems and the great potential of developing insecticide resistance [7,8,9], new methods of insect control are needed. One potential method is the use of ozone, which has proven effective in previous studies [10,11,12,13,14,15]. Although the mechanism of action of ozone on insects is not fully known, the respiratory system of insects is the targeted area of action [16]. Ozone is very reactive and damages the cell membranes of organisms, causing oxidative stress. When ozone enters a cell, it oxidizes all of the essential components (enzymes, proteins, DNA, RNA). As the membrane becomes damaged during this process, the cell decays and the organism dies [17,18,19].

Ozone (O_3_) is a gas composed of three oxygen atoms that naturally occurs in small quantities in the upper atmosphere, where its maximum concentration does not exceed 0.001%. Ozone has a half-life of 20–50 min before rapidly decomposing into diatomic oxygen (O_2_), a natural component in the atmosphere [20]. Decomposition into O_2_ releases one oxygen atom that is highly reactive [21]. This single free oxygen reacts with the cell membranes of organisms and disrupts their normal cellular activity [18]. If ozone is coupled to a volatile organic compound in air, the free oxygen atom reacts with it, removing odors [18]. Since ozone can be easily generated at the point of treatment using only electricity and air, it offers several safety advantages over chemical pesticides. First, there are no repositories of toxic chemicals, nor risks of residual pesticides from chemical mixing or disposal, in addition to no packaging waste being produced [22]. Second, because of the short half-life, it reverts to naturally occurring oxygen, leaving no residue on the product. Third, if necessary, it is possible to neutralize ozone by thermally activated carbon as well as by the catalytic reduction of emissions [22].

For broader use, ozone is defined as a highly reactive and potent oxidizing agent and was classified as GRAS (generally recognized as safe) in 1982 by the US Environmental Protection Agency (US EPA) [23]. In June 2001, the US Food and Drug Administration (US FDA) approved ozone for use as “an antimicrobial agent for the treatment, storage and processing of food in gaseous and aqueous phases”. Ozone has also been approved by the United States Department of Agriculture (USDA) in organic food processing [23] and is well known to the food industry. It has long been used in food processing for disinfection and odor, taste, and color removal [24,25,26,27]. Around the world, ozone is used to purify drinking water, destroy bacteria, freshen up air, and reduce aflatoxin contamination [16,28,29,30,31]. More recently, ozone has been used in agriculture. Potential applications include odor reduction in the poultry industry, the removal of lagoon waste odor from pig breeding, and the reduction of pathogens in the storage of grapes, potatoes, and onions. Apart from a few published papers in the 1960s and 1980s, little is known about the effects of ozone on fungi [32,33,34], and even less is known about the effect of ozone on insects. Prior to 1980, ozone toxicity was investigated in agriculture on insect pests in the field, and there were very few studies on insects in warehouses [11,12,13,14,15]. In recent years, the toxicity of ozone has been investigated on pests of stored goods (silos and containers). Efficacy has been investigated in several laboratory and field studies [3,10,11,12,13,14,15,35,36,37,38]. These studies have shown the toxic effect of ozone on insects in storage. However, there are many inconsistencies in the published reports that are summarized below.

Previous studies have found that ozone treatment can adversely affect storage pests from the order Coleoptera, including maize weevil (*Sitophilus zeamais* (Motsch)), rice weevil (*Sitophilus oryzae* (L.)), and red flour beetle (*Tribolium castaneum* (Herbst)), and from the order Lepidoptera, including Indian meal moth (*Plodia interpunctella* (Hübner)) and Mediterranean flour moth (*Ephestia kuehniella* (Zeller)) [11,12,13,14,15]. Ozone has also shown the potential to suppress phosphine-resistant insect strains [35,36]. Ozone has been shown to be lethal to insects feeding on grain and in grain. The studied insects feeding on grain include beetles, such as the rusty grain beetle (*Cryptolestes ferrugineus* (Stephens)) [37], sawtoothed grain beetle (*Oryzaephilus surinamensis* (L.)) [3,35,37,38], red flour beetle (*Tribolium castaneum* (Herbst)) [3,10,37,38], confused flour beetle (*Tribolium confusum* (J. Duv.)) [3,10,35,38], and drugstore beetle (*Stegobium paniceum* (L.)) [3]; moths, such as the Mediterranean flour moth (*E. kuehniella* (Zell)) [3] and Indian meal moth (*P. interpunctella* (Hübner)) [3,11,37]; and booklice (psocids) [37]. The insects investigated feeding on grain were weevils (*Sitophilus* spp.) [3,11,37], the lesser grain borer (*Rhyzopertha dominica* (F.)) [3,35,37], and the Angoumois grain moth (*Sitotroga cerealella* (Oliv.)) [3]. Studies have shown that the most resistant stage of the insect to ozone is the egg; that is, ozone has no effect on insect eggs [3,13,37]. In these same studies, the relative levels of sensitivity between the larval, pupal, and adult stages differed. For example, Bonjour et al. [37] and Hansen et al. [3] found that the pupal and adult stages were the most sensitive developmental stages of insects to ozone. Leesch [13] found that the pupal stage of *P*. *interpunctella* moths was the most resistant after eggs. However, Erdman [10] investigated the levels of susceptibility of *T*. *castaneum* larva and pupa to ozone and found a decrease in sensitivity with age, that is, younger larval stages and very young pups showed greater sensitivity to ozone application [39]. The ozone dose and exposure time leading to 100% mortality did not differ for larval age in any of the 11 insect species studied [3]. Generally, in all studies, insect mortality was higher when insects were ozonated and left without availability to food. When food was available, ozone had the least effect on adult insects and the greatest effect on larva control [40].

Larger studies on the use of ozone for insect control in Europe have not been conducted, with the exception of one study in Denmark by McDonough et al. [41], and the results of US and Asian investigators concern harmful insects in cereal stores with often contradictory results. This research is the first of its kind in the Republic of Croatia and is among the first in Europe. The question to which we have sought an answer is the following: can ozone application be a new strategy in the control of storage pests?

Prior to carrying out this study, it was presumed that ozone would have a harmful effect on wheat weevil and could be used as a plausible alternative to chemical suppression. The overall objective of the paper was to determine the effectiveness of ozone in the suppression of the wheat weevil. The specific objectives of the work were to determine: (1) the required ozone exposure time to achieve satisfactory wheat weevil mortality (>95%); (2) the effect of ozone on the walking activity and speed or velocity of insects; and (3) the relationship between mortality and time after ozone treatment.

## 2. Materials and Methods

This study was carried out in the laboratory of the Department of Agricultural Zoology, Faculty of Agriculture, University of Zagreb, Croatia, in the period from January to March 2019. The controlled laboratory temperature was 24 ± 2 °C, and the humidity was 50–60% throughout the study period.

### 2.1. Ozonator

An ozonator from the Tiens d.o.o. model DiCHO (Tiens Company, Beijing, China) was used for ozonation. This device uses an electrical charge to transform O_2_ from the air to O_3_—ozone. The ozone output was 150 mg/h [42]. This ozone output generated 2.5 mg of ozone/L of water or 0.001 ppm of ozone/L of air [42]. It is important to note that this is a quantity that is naturally present in the higher layers of the atmosphere and is completely non-harmful to humans (0.040–0.100 ppm is the maximum allowable concentration for use in human presence) [19]. As a very small ozonator of limited use was employed (primarily registered for food and water disinfection [42]), the ozone output could not be changed, and the time of exposure to ozone also defined the amount of ozone used as a test variable.

### 2.2. Insects and Ozonation

Wheat weevil adults (*S. granarius*) collected in stored maize in a private storage facility were used in this study. Ozonation was carried out in a purpose-built plastic chamber of 20 × 20 cm. An opening was made at the top of the chamber through which an ozonator tube for ozone release was introduced. The wheat weevils were ozonated in two ways: (i) directly onto weevils alone and (ii) weevils mixed in with grain. The independent variable in the experiment was exposure time to ozone (10, 20, 30, 60, or 120 min); the dependent variable was weevil mortality. Both treatments had the same variable. Each condition was examined with four repetitions, and each repetition comprised 20 wheat weevil individuals. The containers in which the insects were ozonated were made of glass, 50 mL in volume, and without a lid so that ozone could flow freely throughout the chamber. Each method of treatment had a corresponding control set that was not exposed to ozone. A total of 880 weevils were included in the study (400 directly treated weevils, 400 weevils treated in grain, 80 non-treated/control weevils). A grain of corn was added to the container after direct ozonation of weevils alone, to ensure the same conditions for all samples. Mortality for all repetitions of all variants was recorded on the first, second, third, seventh, tenth, and fifteenth day after ozonation. In addition to mortality, the walking response of the surviving individuals was monitored in each trial term in such a way that the average distance traveled by the insect individuals over a period of 20 s was measured using gridded paper (millimeters). In addition to mobility, insect velocity was monitored throughout the trial by drawing a 20 cm circle on gridded paper (millimeters). Insects from each repetition were placed in the center, and the time required for the majority of the surviving population to step outside the circle was measured. Table 1 shows the experimental approach taken.

### 2.3. Data Analysis

Insect mortality in each replication of each variant was recorded throughout the study period, and ozone efficiency was calculated based on the percentage of mortality using the Abbot formula [43]. The efficacies of all variants were subjected to analysis of variance (ANOVA) to determine the difference in the effect of ozone on insects at different exposure times. In order to stabilize the variance where appropriate, data were transformed by the use of *log* (*x* + 1) or *arc sin*
$\sqrt{x}\;$. ANOVA was performed on efficacy data, insect velocity, and mobility data to determine the effect of ozone on the above parameters. A post-hoc means test was used when significant differences were found (Tukey’s HSD). To determine the relative toxicity of the ozone treatment applied directly to insects and insects in grain, efficacy data were analyzed by probit regression analysis in order to transform the sigmoid dose–response curve to a straight line. For each group of insects, the curve was used to inversely predict the dose corresponding to a 50% reduction in survival (LC_50_). Statistical data processing (ANOVA, Tukey’s HSD test, Probit analysis) was performed using ARM 2019^®^ GDM software (Gylling Data Management, Inc., Brookings, SD, USA) [44].

## 3. Results

### 3.1. Efficiency of Ozone in Wheat Weevil Suppression

The duration of ozonation and the ozonation type (directly to insects or insects in grain) significantly affected insect mortality as well as the elapsed time since ozonation (Table 2). The highest efficacy was observed in variants treated for 120 min. Shorter ozone exposure did not result in satisfactory efficacy, and no significant difference in efficacy was observed between variants ozonated for time periods shorter than 120 min. The wheat weevils in grain had a lower mortality rate than directly treated weevils. A 100% mortality rate for directly treated weevils was observed in 120 min 7 days after ozonation. The maximum (and satisfactory efficacy, i.e., >95%) mortality was only established on the 15th day after ozonation for 120 min in the insects ozonated in grain treatment.

### 3.2. The Influence of Ozone on the Walking Response and Velocity of Wheat Weevils

In all treatments, a lower walking response was observed on the first and second days after ozonation, after which walking increased and ultimately decreased significantly on the 15th day of the experiment (Table 3). The greatest differences were observed between the weevils treated for 120 min. In the 120 min treatment of directly ozonated insects, the weevils were mobile on the second and third days, but significantly less than weevils from the other treatments. In addition, 100% mortality was achieved in this group by the seventh day of the experiment. In the treatment of ozonated insects in grain, the 120 min time tested ones showed a significantly lower walking response on the second day after ozonation, and their response remained the lowest until the end of the experiment. By analyzing each treatment group separately (directly to insects/insects in grain), it was evident that ozone exposure for 120 min had a negative effect on all weevils, and the lowest walking response was measured in insects from this treatment.

The velocity tests showed that the weevils in all treatments were faster immediately after ozonation (the first day of reading), while the velocities decreased in the days after ozonation (Table 4). Ozone exposure for 120 min had a negative effect on all wheat weevils, and the lowest velocity of weevils was recorded in these variants. The lowest velocity of the weevils was recorded in variants where the insects were directly ozonated for 120 min. If experiments are considered where only the weevils were treated with grain, the velocity was significantly lower in the 120 min treatment group compared to the other groups for the whole experimental period. There were no significant differences between the other experiments in both treatments (directly to insects/insects in grains) over the entire testing period.

Calculated LC_50_ across 15 days gradually decreased. The lowest LC_50_ was calculated for both insect groups 15 days after ozonation. However, LC_50_ was lower when ozone was applied directly to insects than if ozone was applied to insects in grain treatment. The slope of the regression lines for the group of insects directly treated varied between 2.13 and 3.6, while the slope for the group of insects treated in grain varied from 1.17 to 2.22 (Table 5).

## 4. Discussion

This study was conducted with the main aim of determining the effectiveness of ozone in the suppression of wheat weevils and to establish the duration of ozonation required to cause mortality as well as the impact of ozone on physical parameters of the weevils themselves (i.e., walking response and velocity). This study demonstrated that ozone had a negative effect on adult wheat weevils causing up to 100% mortality; this is the first study to demonstrate this result. These important findings complement the results of previous studies where the effectiveness of ozone for other storage pests, including maize weevil, rice weevil, and red flour beetle, was shown [3,11,12,13,14,15,37].

As expected, in this study, a higher efficacy was observed in treatments in which insects were directly exposed to ozone as compared to treatments where weevils were ozonated together with grain. Besides the higher efficacy and lower LC_50_ when ozone was applied directly to insects comparing to treatment of insects in grain, this study confirmed that the regression slopes were higher if ozone was applied directly to insects. This means that the differences in LC_95_ would be even higher. The ozone levels investigated in the abovementioned studies were 20, 35, 75, 100, and 135 ppm—several times higher than the doses used in this study (in this study, the maximum amount of applied ozone was 0.006 ppm). For three *Sitophilus* species (not *S. granarius*), Hansen et al. [3] reported 100% mortality in the application of 35 ppm ozone applied over five days to insects only and in the application of 135 ppm ozone applied over eight days to insects ozonated together with grain. McDonough et al. [41] determined 1800 ppm of ozone to be the required level for achieving 100% mortality in *S. zeamais*. In this study, the maximum amount of ozone was 0.002 ppm, equaling 300 mg (5 mg/L) of the output power in an ozonation time of two hours and resulting in 100% mortality of wheat weevils within one week after ozonation. The level of used ozone was much lower than that tested in previous studies. The large observed differences in the amount of ozone required to achieve high efficiency can be attributed to the non-detailed descriptions of the ozonators used in other studies, as well as the unknown method of calculating the amount of ozone in a volume of one liter of air. As the aforementioned studies [35,36,37,38,39,40,41] used commercial ozonators with far higher power outputs, and due to the very high toxicity of ozone for humans in quantities above 0.1 ppm, we assume that the amounts presented in these studies actually corresponded to the ozone output power, rather than that calculated in air. However, the ozone levels and the duration of ozonation used in this study were much smaller and shorter than in previous studies [35,36,37,38,39,40,41]. Thus, a confirmation of these experimental results should be scaled up and conducted in storage systems under field conditions. Our results also suggest that Croatian wheat weevil populations are highly sensitive to ozone fumigation. This was expected since the investigated weevil populations were from private storage facilities that had not been treated with any prior pest control products.

In addition to efficiency, the effect of ozone on the walking and velocity of wheat weevils were investigated. The only previous walking response and velocity study conducted on the maize weevil was by Sousa et al. [45]. These authors found that walking response decreased as ozone efficiency increased. As expected, this was confirmed by the results of our study, and it can be concluded that ozone impacts on the walking response and velocity of treated insects, factors that have not been studied for any other insect so far. Other interesting results from our work include a decrease in walking activity shortly after ozonation, an increase in activity after a few days, and a recurrent decrease in walking activity found about ten days after ozonation. As expected, the velocity of wheat weevils decreased after ozonation, following the same trend throughout the entire investigation period. A significant reduction in walking activity and velocity were caused by ozonation for 120 min, which was the maximum investigated time for the effect of ozone on these parameters. The ability of ozone to reduce the walking activity and velocity of treated insects is a positive feature in pest control in storage systems, thereby reducing the possibility of insects escaping from treated objects [45]. As the type of storage may also have an impact on the walking activity and velocity of the insects, this research needs to be conducted in different storage systems under field conditions in order to provide a more detailed understanding of ozone as a potential eco-friendly fumigant for the control of storage pests.

## 5. Conclusions

The experiments conducted in this study demonstrate the effectiveness of ozone in suppressing wheat weevils. As the ozonator used in this study was for small (domestic) application and it was not possible to change the amount of ozone applied in the specific time frame, we cannot make a conclusion as to whether the mortality rate would be affected by using a greater amount of ozone or a longer period of exposure. However, ozone has the potential for use in storage pest control because: (i) it can be generated at the site of application, (ii) it does not leave toxic residue after application, and (iii) the risks to pest management personnel are minimal due to its rapid degradation. Together, the results obtained represent new data for the control of weevils in the Republic of Croatia (and the European Union) and will contribute to the overall knowledge of the use of this gas in agriculture. Ozonation has the potential to become a realistic choice for the control of harmful organisms in the storage systems of raw materials for human and animal consumption or final food products, alone or as a complement to other methods. This is method of insect control in grain storage especially warrants further investigation because it is an eco-friendly alternative to chemical control.

## Figures and Tables

**Table 1 insects-10-00357-t001:** Applied ozone according to variant.

Variants in the Experiment (Ozone Exposure in Minutes)	The Total Amount of Ozone Per Variant (mg)	The Applied Concentration of Ozone in Air (ppm)
10	25	0.0002
20	50	0.0003
30	75	0.0005
60	150	0.0010
120	300	0.0020

**Table 2 insects-10-00357-t002:** Ozone efficiency (%) in wheat weevil mortality.

OzonationType	Duration of Ozonation (min)	Days after Ozonation					
		1	2	3	7	10	15
Directly to insects	10	0.0 ± 0.0 c *	0.0 ± 0.0 c	5.3 ± 0.4 c	13.5 ± 0.1 c	25.9 ± 1.5 bc	58.9 ± 26.7 a–d
	20	0.0 ± 0.0 c	0.3 ± 0.8 bc	11.4 ± 9.4 bc	20.9 ± 0.3 c	36.6 ± 2.9 bc	59.6 ± 28.8 a–d
	30	0.6 ± 6.5 bc	0.7 ± 1.3 bc	1.9 ± 9.45 c	6.6 ± 0.5 c	21.5 ± 2.1 bc	45.6 ± 6.4 bcd
	60	6.1 ± 11.3 abc	11.7 ± 0.3 ab	12.8 ± 3.8 bc	42.1 ± 0.2 bc	67.5 ± 1.5 ab	88.2 ± 16.6 abc
	120	17.3 ± 8.9 a	28.5 ± 0.9 a	69.8 ± 8.1 a	100.0 ± 0.0 a	100.0 ± 0.0 a	100.0 ± 0.0 a
Insects in grain	10	0.0 ± 0.0 c	2.1 ± 1.3 bc	2.1 ± 9.9 c	7.2 ± 0.5 c	8.7 ± 0.7 c	20.7 ± 14.3 cd
	20	0.0 ± 0.0 c	0.7 ± 1.4 bc	2.0 ± 9.7 c	9.5 ± 0.2 c	14.2 ± 1.1 c	17.0 ± 10.3 d
	30	0.0 ± 0.0 c	1.3 ± 1.0 bc	1.3 ± 7.7 c	11.0 ± 0.2 c	12.7 ± 1.2 c	13.8 ± 9.4 d
	60	2.4 ± 12.7 abc	6.3 ± 1.6 abc	14.9 ± 15.1 bc	24.3 ± 0.5 c	31.3 ± 2.7 bc	44.2 ± 33.2 bcd
	120	7.8 ± 13.4 ab	24.2 ± 1.1 a	41.3 ± 9.6 ab	76.9 ± 0.2 ab	94.5 ± 0.4 a	97.8 ± 10.1 ab
HSD ** *p* = 0.05		11.5	14.9	22.8	38.4	48.0	50.4

* Mean values of the same column followed by the same letter are not significantly different (*p* ≥ 0.05; HSD test); ** HSD was determined by comparing the ozone efficiency between the duration of ozonation for each checking time (days after ozonation).

**Table 3 insects-10-00357-t003:** Walking response (mm in 20 s) of *Sitophilus granarius* after ozone exposure.

Ozonation Type	Duration of Ozonation (min)	Days after Ozonation					
		1	2	3	7	10	15
Directly to insects	10	45.5 ± 5.1 ab *	45.0 ± 15.8 abc	76.8 ± 5.8 ab	65.0 ± 24.3 a	24.3 a ± 24.3 a	44.0 ± 20.1 a
	20	64.8 ± 14.2 a	67.8 ± 18.0 a	84.0 ± 16.2 a	69.5 ± 4.5 a	4.5 a ± 4.5 a	23.8 ± 24.9 ab
	30	40.5 ± 17.2 ab	63.0 ± 17.2 ab	71.0 ± 11.7 ab	48.3 ± 17.0 ab	17.0 ab ± 17.0 ab	16.5 ± 14.0 ab
	60	59.8 ± 26.6 a	56.8 ± 1.7 abc	54.5 ± 18.1 abc	51.3 ± 13.5 ab	13.5 ab ± 13.5 ab	17.3 ± 25.5 ab
	120	0.0 ± 0.0 b	17.5 ± 35.0 c	12.5 ± 25.0 c	0.0 ± 0.0 c	0.0 c ± 0.0 c	0.0 ± 0.0 b
Insects in grain	10	45.0 ± 13.3 ab	36.8 ± 10.9 abc	43.8 ± 2.4 abc	46.0 ± 3.8 ab	44.5 ± 10.4 ab	33.3 ± 6.2 ab
	20	35.8 ± 11.6 ab	44.3 ± 8.4 abc	39.0 ± 28.7 abc	42.3 ± 13.1 ab	43.5 ± 11.6 ab	31.5 ± 14.6 ab
	30	42.5 ± 20.2 ab	32.0 ± 4.8 abc	40.8 ± 13.9 abc	48.0 ± 17.6 ab	44.8 ± 11.5 ab	41.3 ± 11.4 a
	60	49.3 ± 16.9 a	57.3 ± 14.8 abc	46.8 ± 16.1 abc	48.3 ± 15.5 ab	42.8 ± 28.7 ab	27.3 ± 20.0 ab
	120	51.0 ± 39.7 a	23.8 ± 28.7 bc	30.8 ± 36.7 bc	16.8 ± 18.0 c	7.8 ± 9.0 b	0.0 ± 0.0 b
HSD ** *p* = 0.05		46.1	41.8	0.4	35.2	44.8	41.2

* Mean values of the same column followed by the same letter are not significantly different (*p* ≥ 0.05; HSD test); ** HSD was determined by comparing the ozone efficiency between the duration of ozonation for each assessment time (days after ozonation).

**Table 4 insects-10-00357-t004:** Velocity (sec) of *Sitophilus granarius* after ozone exposure.

Ozonation Type	Duration of Ozonation (min)	Days after Ozonation					
		1	2	3	7	10	15
Directly to insects	10	155.0 ± 5.8 ab *	137.5 ± 9.6 ns	9.6 ns ± 33.2 ab	117.5 ± 71.8 a	117.5 ± 79.3 ab	90.0 ± 29.4 ab
	20	125.0 ± 19.2 abc	115.0 ± 33.2 ns	33.2 ns ± 56.6 abc	77.5 ± 55.6 ab	77.5 ± 47.9 abc	107.5 ± 99.1 ab
	30	142.5 ± 9.6 ab	120.0 ± 21.6 ns	21.6 ns ± 9.6 ab	112.5 ± 17.1 a	112.5 ± 18.3 abc	90.0 ± 57.2 ab
	60	107.5 ± 9.6 bc	105.0 ± 23.8 ns	23.8 ns ± 25.0 abc	100.0 ± 43.2 ab	100.0 ± 49.9 abc	55.0 ± 65.6 ab
	120	82.5 ± 35.0 c	85.0 ± 20.8 ns	20.8 ns ± 42.7 c	0.0 ± 0.0 b	0.0 ± 0.0 c	0.0 ± 0.0 b
Insects in grain	10	155.0 ± 12.9 ab	152.5 ± 28.7 ns	155.0 ± 12.9 a	140.0 ± 8.2 a	135.0 ± 23.8 ab	127.5 ± 45.7 ab
	20	165.0 ± 17.3 a	157.5 ± 17.1 ns	160.0 ± 16.3 a	157.5 ± 22.2 a	172.5 ± 22.2 a	147.5 ± 46.5 a
	30	160.0 ± 14.1 ab	145.0 ± 10.0 ns	127.5 ± 17.0 ab	120.0 ± 20.0	170.0 ± 8.2 a	150.0 ± 24.5 a
	60	140.0 ± 40.0 ab	115.0 ± 63.5 ns	112.5 ± 68.5 abc	97.5 ± 63.4 ab	102.5 ± 77.2 abc	105.0 ± 71.4 ab
	120	107.5 ± 37.8 bc	100.0 ± 59.4 ns	57.5 ± 40.3 bc	70.0 ± 54.8 ab	35.0 ± 41.2 bc	5.0 ± 5.8 d
HSD ** *p* = 0.05		56.5	80.7	91.8	107.8	113.8	132.2

* Mean values of the same column followed by the same letter are not significantly different (*p* ≥ 0.05; HSD test); ** HSD was determined by comparing the ozone efficiency between the duration of ozonation for each assessment time (days after ozonation).

**Table 5 insects-10-00357-t005:** Comparisons of probit analysis results of ozone efficiency data.

Days after Ozonation	Ozonation Type	LC_50_	95% Confidence Limits	Regression Line Equation
1	Directly to insects	701.69	544.58–1037.22	Y = −1.0622 + 2.1300 X
	Insects in grain	1032.21	721.94–1883.66	Y = −0.9277 + 1.9669 X
2	Directly to insects	528.69	444.82–664.74	Y = −1.0594 + 2.2251 X
	Insects in grain	1505.63	993.30–2716.34	Y = 1.2821 + 1.1700 X
3	Directly to insects	255.59	234.33–281.96	Y = −1.3817 + 2.6507 X
	Insects in grain	477.68	400.71–593.03	Y = 0.5588 + 1.6577 X
7	Directly to insects	210.48	184.72–249.95	Y = −1.0686 + 2.6121 X
	Insects in grain	206.81	188.42–229.58	Y = 0.1640 + 2.0885 X
10	Directly to insects	127.66	118.91–138.69	Y = −1.1763 + 2.9326 X
	Insects in grain	161.46	147.53–178.79	Y = 0.0981 + 2.2200 X
15	Directly to insects	101.02	95.84–106.56	Y = −2.2073 + 3.5958 X
	Insects in grain	146.91	130.43–168.97	Y = 1.6111 + 1.5638 X

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
