# Peer review of "Ozone Effectiveness on Wheat Weevil Suppression: Preliminary Research"

_insects, 2019, doi:10.3390/insects10100357_

Round 1
Reviewer 1 Report
Do you have data on pupal, eggs and larva? Adults are relatively easy to kill. the challenge is in the other life stages and those life stages are the most damaging to grain.
More testing is needed. This research gives what could be considered "preliminary research" that would justify the need to continue with more intensive research.
Author Response
Reviewer #1:
Q1) Do you have data on pupal, eggs and larva? Adults are relatively easy to kill. the challenge is in the other life stages and those life stages are the most damaging to grain.
More testing is needed. This research gives what could be considered "preliminary research" that would justify the need to continue with more intensive research.
Response: In this stage of investigation we only apply ozone on adults. The reason is that adults are the main stage that are inspected in storage systems in our country, and considering thresholds for adults the companies make decisions about suppression. For future work we will include all stages of pests (of different species). In this paper we added Preliminary research in title to make it clear that more intensive research will be done in future.
Reviewer 2 Report
This revised manuscript is greatly improved, but more improvement is needed
The English grammar is still not acceptable, more editing is required
The Tables 2-4 are not acceptable. Put the means and the standard errors and the mean separation letter on one line. Use one significant digit for the averages. Put the Tables in landscape mode not portrait. Example 23.1±2.2, not 23.12±2.23. It is not clear if means are compared across days or down in dosages. All three Tables must be re-arranged.
It is P greater than or equal to 0.05, not P> 0.05
It is walking response, not waking response. There are several instances of this misspelling error. See line 20 in abstract for the first one.
Lines 135-136, Do not have one-line paragraphs!
Lines 232-234, again do not have short one-sentence paragraphs
Author Response
Reviewer #2:
This revised manuscript is greatly improved, but more improvement is needed
Q1) The English grammar is still not acceptable, more editing is required
Response: Manuscript has been editing by MDPI editing service. Also, Prof. K. Mikac (University of Wollongong, Australia), as English native speaker, edit and improved out manuscript. Is now added in Acknowledgment.
Q2) The Tables 2-4 are not acceptable. Put the means and the standard errors and the mean separation letter on one line. Use one significant digit for the averages. Put the Tables in landscape mode not portrait.
Example 23.1±2.2, not 23.12±2.23. It is not clear if means are compared across days or down in dosages. All three Tables must be re-arranged.
Response: All tables have been re-arranged according reviewer comments.
Q3) It is P greater than or equal to 0.05, not P> 0.05
Response: done
Q4) It is walking response, not waking response. There are several instances of this misspelling error. See line 20 in abstract for the first one.
Response: All manuscript has been edited by native speaker.
Q5) Lines 135-136, Do not have one-line paragraphs!
Response: corrected
Q6) Lines 232-234, again do not have short one-sentence paragraphs
Response: corrected
Round 2
Reviewer 1 Report
Line 11: change "9-20%" to just "20%".
Author Response
The authors thank reviewer for comments which improved greatly our manuscript.
This manuscript is a resubmission of an earlier submission. The following is a list of the peer review reports and author responses from that submission.
Round 1
Reviewer 1 Report
This manuscript is not acceptable as submitted because for all the Tables, there is no estimate of variation with the means. You cannot publish Tables stating that means are significantly different without showing the variation about those means. Use standard error, and revise Tables accordingly
Much more detail is needed regarding the statistical analysis. You need to completely describe what tests of significance you are using under the statistical package.
Statisticians were discouraging the use of Duncan's Multiple Range Test 20 years ago because it did not account for experiment wise error. Use a different test.
You have two ordered sequences of concentration and time. Most authors would use regression analysis to determine significance between these ordered sequences, not a mean separation test.
The overall grammar, construction, and interpretation seems fine, but the above must be addressed if the authors wish to revise the paper.
Reviewer 2 Report
This manuscript is well written. See the attached file for minor changes to the paper. The work presented may provide a very early baseline for research that would be more impactful. It is not new knowledge that ozone kills insects. Taking this information into a grain storage environment is where the challenges present themselves. Perhaps the authors would consider this next stage of research. That would be helpful to this scientific community.

Reviewer 3 Report
This study contributes nothing except that they used the Sitophilus granaries as a test subject. There are three common weevils in stored grain. The maize weevil and rice weevil have been tested and published in literatures. This study confirmed the conclusion published in the literature.
This study also conducted minimum tests (the same ozone production rate with different treatment times) due to their equipment limitation. However, the author should test larval, pupa, and eggs.
Considering this minimum tests and minimum contribution (which has been concluded in the literature), I would suggest this paper not published as a full article, but a short communication, or re-submit it until the completion of larval, pupa, and egg.
The conclusions have many the same statements as in the abstract and in the introduction. Also, the statements from line 8 to 17 should not be in the abstract because all these statements are the common knowledge and were not discovered in this study. If Line 8 to 17 are deleted, the abstract will have less 110 words. This short abstract is caused by the minimum study and contribution.